# Design and Molding of Thyroid Cartilage Prosthesis Based on 3D Printing Technology

**Guoqing Zhang** [1,*] **, Junxin Li** [1] **, Chengguang Zhang** [1] **and Anmin Wang** [2]

1   School of Mechanical and Electrical Engineering, Zhoukou Normal University, Zhoukou 466000, China;
    lijunxin1995@163.com (J.L.); zchengguang2022@163.com (C.Z.)
2   School of Mechanical and Automotive Engineering, South China University of Technology,
    Guangzhou 510640, China; wanganminhnlg@163.com
*   Correspondence: zhangguoqing1202@sohu.com

**Abstract:** The modeling efficiency, matching, and biocompatibility are key factors affecting the surgical success of a personalized thyroid cartilage prosthesis. We performed three-dimensional reconstruction of a thyroid cartilage prosthesis by combining reverse and forward methods, and then completed the prosthesis design with total or partial resection using a parametric modeling method. Direct manufacturing was performed using selective laser melting (SLM) molding equipment and TC4 material. The structure of the completed implant unit was optimized. The results show good modeling effects for the thyroid cartilage prosthesis with either total or partial resection by the parametric modeling method. Good matching performance was achieved, with overlap suspension between the pillars that meets the requirements of SLM manufacturing. Additionally, the use of SLM molding to produce the thyroid cartilage prosthesis resulted in less powder adhesion on the surface and no obvious nodulation between the porous pillars, allowing the direct use of the prosthesis after simple post-treatment. Overall, these results should facilitate the direct application of personalized implants.

**Keywords:** selective laser melting; parametric modeling; truss structure; pore structure; layout

## 1. Introduction

Patients with laryngeal carcinoma or larynx injury will inevitably require the surgi-cal removal of part or all of the thyroid cartilage. Because the thyroid cartilage is non-renewable tissue, a defect will seriously affect patient swallowing, phonation, and respiration, reducing the quality of life after surgery and potentially resulting in death. With the development of human bone repair and replacement surgery, thyroid cartilage repair can be performed as a strategy to relieve patient pain, restore the function of thyroid cartilage, and improve patient quality of life [1]. In traditional surgery to repair thyroid cartilage, a prosthesis is usually prepared by manual shaping of flat metal mesh. The ability to effectively shape the implant completely depends on the skills and experience of the doctor, and the resulting implant may exhibit a variable degree of matching with the remaining part of the patient's natural thyroid cartilage. Although the side effects caused by cartilage loss are effectively overcome after the surgery, the appearance of the cartilage cannot be restored, and biocompatibility may be poor. A personalized prosthesis that is partially or completely porous is lightweight and offers good biocompatibility, but there are challenges for its design and manufacture.

The structural parameters such as porosity, surface-volume ratio, and average pore diameter can be adjusted by adjusting the input parameters of the porous structure designed by parametric modeling technology. Additive manufacturing technology (3D printing) uses special data processing software to slice and layer a 3D model, obtain section data, import the data to the 3D printing equipment, and then manufacture the solid parts by layer manufacturing. With layer manufacturing, 3D printing technology can produce nearly

any geometric part, with advantages of single piece work, small batch, complex geometry structure, and dense tissue after processing [2], allowing its use for the direct manufacturing of a thyroid cartilage prosthesis with biological fixation. In particular, selective laser melting (SLM) technology is a 3D printing technology based on laser-melted metal powder [3–5].

Wang et al. [6] compared use of transplanted tissue repair and wound repair methods for treatment of thyroid cartilage defects, and found no significant differences between the two groups for long-term effects, survival rate, and the rate of casing withdrawal. Liu et al. [7] used bone morphogenetic protein complex to repair defects of laryngeal thyroid cartilage in situ using CAD and CAM technology, with good repair effect and good osteoinduction and biocompatibility. Feng et al. [8] examined the therapeutic effect and application value of two kinds of repair methods for the reconstruction of laryngeal function after partial laryngectomy for laryngeal carcinoma, finding that relay muscle thyroid cartilage outer perichondrium flaps and ventricular zone downward approaches could well reconstruct the shape of the laryngeal cavity and restore the function of the thyroid cartilage. Wang et al. [9] prepared HA foam ceramics with a foaming method to repair thyroid cartilage injury. Microscopic observation showed new bone growth and good pronunciation and eating behaviors for patients with the repaired thyroid cartilage. Tian et al. [10] applied digital combined 3D printing technology as an auxiliary approach for the excision of lesions and reconstruction of residual larynxes, which significantly reduced patient hospital stays, shortened the recovery cycle, and improved the quality of life compared with CHEP surgery. Zhang et al. [11] prepared a thyroid cartilage scaffold with different hydroxyapatite (HA) contents (volume fraction of 60%, 50%, 40%) through foam gel injection molding, which provided a new scaffold material for the repair and reconstruction of thyroid cartilage defects. Bhavana et al. [12] used the Crista iliaca as a graft to repair defects of the laryngotracheal cartilage and restore function. However, there is limited autogenous cartilage, and this approach will present new trauma to the patient during transplantation, with potential postoperative complications, leading to the failure of the surgery. Hallak et al. [13] reported the case of thyroid cartilage fracture under mini bone fracture plate repair and dislocation.

Autogenous bone grafting and foaming methods are the main technologies applied for the repair of thyroid cartilage. However, limitations include the source of autogenous cartilage and the difficulty of controlling the pores in foam ceramics by foaming methods. The development of SLM molding technology and parametric modeling technology has provided the possibility for solving the above problems.

## 2. Materials and Methods

### 2.1. Implant Design Requirements

The design goal of personalized implants is to meet the requirements of implantation and allow the repair and reconstruction of human tissues or organs. Therefore, mechanical properties, biocompatibility, manufacturability, and matching with peripheral bone tissues must be considered in the design of a prosthesis.

The biological performance requirements of a thyroid cartilage prosthesis are as follows [14–18]: the optimal pore size range of porous structure for the growth of bone cells is about 100–1000 mm, and a porosity of 50–90% can simulate cancellous bone structure, which is most conducive to the growth of new bone. The larger the surface area volume ratio of the porous implant, the larger the contact area between the porous implant surface and the bone, so the greater the mechanical stimulation of new bone.

There are several mechanical requirements of a thyroid cartilage prosthesis [19,20]. The thyroid cartilage prosthesis should not deform and damage after implantation, so it should have good stiffness and strength. The elastic modulus of the thyroid cartilage prosthesis should be equivalent to the peripheral bone tissue to avoid "stress shielding," which can prevent stimulation of growth of the peripheral tissue. Finally, to improve the impact resistance of the thyroid cartilage, the prosthesis should have energy absorptive capacity.

The thyroid cartilage prosthesis must match the peripheral bone tissue. An implant with good matching can better stimulate the growth of peripheral bone tissue and avoid osteolysis. Additionally, an implant with good matching can reduce the possibility of loosening, making the force of the implant more uniform and improving the service life of the implant.

There are several requirements for prosthesis machinability. The prosthesis should meet the design criteria of parts prepared by SLM molding, such as avoiding sharp corners and thin walls, and having the suspension between parts and the molding datum plane greater than 45°. Additionally, the minimum aperture size should be greater than the limit of laser molding.

### 2.2. Materials and Methods

Based on its high specific strength and good biocompatibility, $Ti_6A_{14}V$ alloy was selected as the raw material to prepare the thyroid cartilage prosthesis. The $Ti_6A_{14}V$ alloy metal powder was produced by Jiangsu Wuxi Falcontech Manufacturing Technology Co., Ltd. (Wuxi, China), and its composition met the requirements of ASTM F136 and GB/T 13810-2007, as listed in Table 1. The powder was prepared by gas atomization and was spherical, as shown in Figure 1. The apparent density $\rho_s$ is 2.55 g/cm$^3$, and the particle size distribution shows a concentrated distribution, with 90% approximately 22 μm, and D50 approximately 28.50 μm.

**Table 1.** The comparison of powder materials manufactured by SLM and ASTM F136 standards.

| Element | $Ti_6A_{14}V$ Powder | ASTM F75 Standard | Element | $Ti_6A_{14}V$ Powder | ASTM F75 Standard |
|---------|----------------------|-------------------|---------|----------------------|-------------------|
| Al | 5.5–6.5% | 5.5–6.5% | N | 0.03% | <0.05% |
| V | 3.5–4.5% | 3.5–4.5% | H | 0.012% | <0.012% |
| Fe | 0.25% | <0.3% | O | <0.08% | <0.13% |
| C | 0.08% | <0.08% | Ti | Balance | Balance |

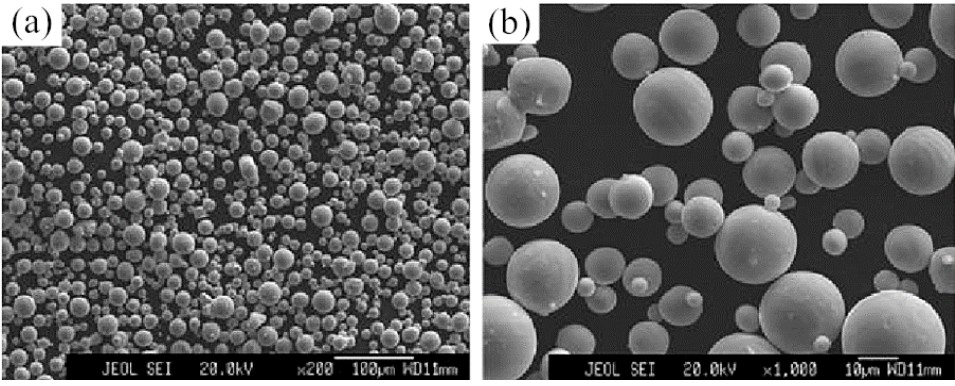

**Figure 1.** Microstructure of $Ti_6A_{14}V$ alloy powder: (**a**) 100× microstructure; (**b**) 1000× microstructure.

GYD 150 molding equipment produced by Shenzhen Sunshine Laser & Electronics Technology Co., Ltd. (Shenzhen, China) was used as the molding equipment. $N_2$ was used as the protective gas, and the oxygen content was controlled below 0.03%. The processing laser power was 180 W, scanning speed was 500 mm/s, the hatch spacing was 80 μm, the processing layer thickness was 25 μm, and X-Y interlaminar scanning strategy was adopted [21]. Three thyroid cartilage prostheses were processed, designed, and completed.

### 2.3. Analysis Methods

A VHX-5000 three-dimensional hyperfocal microscope produced by Japan Keyemce Company (Osaka, Japan) was used to observe the surface morphology of porous parts prepared by SLM molding. A 3D printer Z-603S produced by Aurora Technology Co., Ltd.

(Singapore) was used to print the unresected part of the thyroid cartilage, and then the unresected part of the thyroid cartilage and the resected part of the implant made by SLM molding were assembled to assess matching.

## 3. Results and Discussion

The schematic diagram of the components of the thyroid cartilage is shown in Figure 2. The prominentia laryngea is the most common site of laryngeal carcinoma or larynx injury. According to the amount of prominentia laryngea resection, total resection or partial resection may be required for thyroid cartilage repair. According to the needed thyroid cartilage repair, solid or porous cartilage implants can be designed.

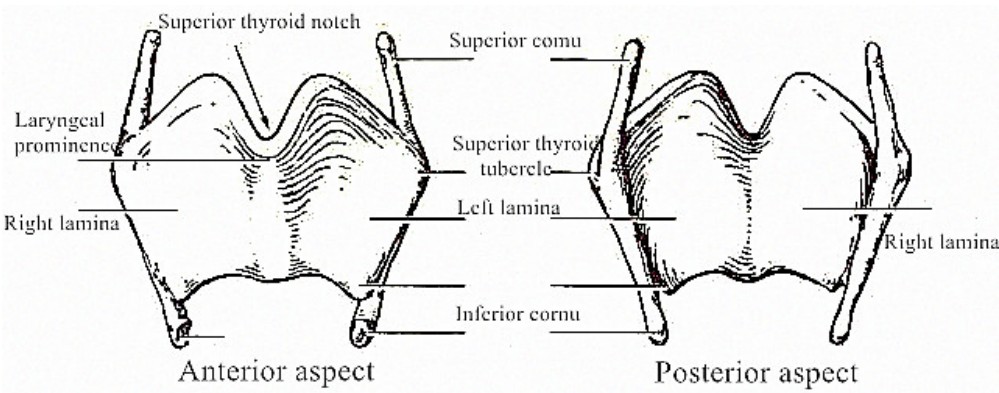

**Figure 2.** Thyroid cartilage.

### 3.1. Design of Solid Thyroid Cartilage Prosthesis

In the design process, achieving good matching between the thyroid cartilage prosthesis and the unaffected sites is a significant challenge. The thyroid cartilage has a complex geometric shape. A high degree of fit between the thyroid cartilage prosthesis and the peripheral bone tissue is required to reduce the possibility of loosening after implantation and improve biocompatibility. The best way to obtain a thyroid cartilage prosthesis with good matching to peripheral bone tissue is to obtain an image of the affected sites and then carry out reverse reconstruction and simulation of osteotomy.

The specific modeling steps are as follows:

(1) First, CT or NMR imaging technologies are used to scan the affected sites of patients with thyroid chondropathy to obtain the image of the affected sites, as shown in Figure 3a. Then, slice images completed by CT scanning are imported into Mimics. During the import process, the correct orientation can be selected in the change orientation window using the right button.

(2) In Mimics, threshold segmentation is applied to the slice image of patients with thyroid chondropathy, and then areas of segmentation that are not connected with each other on the initial threshold segmentation mask are further subdivided into subgroups to generate a new mask. The soft tissue is marked as the starting point, and the end point is marked after the line passes through the bone, creating a strong interface. The raised part represents the threshold value.

(3) After threshold segmentation, the slice image is processed by morphology manipulation, and small burrs on the boundary of the segmentation mask are removed by the open operation (first corrosion and then expansion). The binary image is segmented to remove floating pixels. The 3D reconstruction of the CT model is completed using Calculate 3D models, as shown in Figure 3b. Finally, an optimized thyroid cartilage model is obtained by smoothing and denoising the model of the reconstructed thyroid cartilage prosthesis, as shown in Figure 3c.

(4) The optimized thyroid cartilage model after three-dimensional reconstruction is then imported into Geomagic Studio 2020 for repair, and the reconstructed thyroid cartilage repair model is obtained after griddoctor repair, denoising, smoothing, and substantiation,

as shown in Figure 3d. As shown in Figure 3d, the model of thyroid cartilage prosthesis established by the reverse design method has good modeling effects and can achieve a good fit with the peripheral bone tissue.

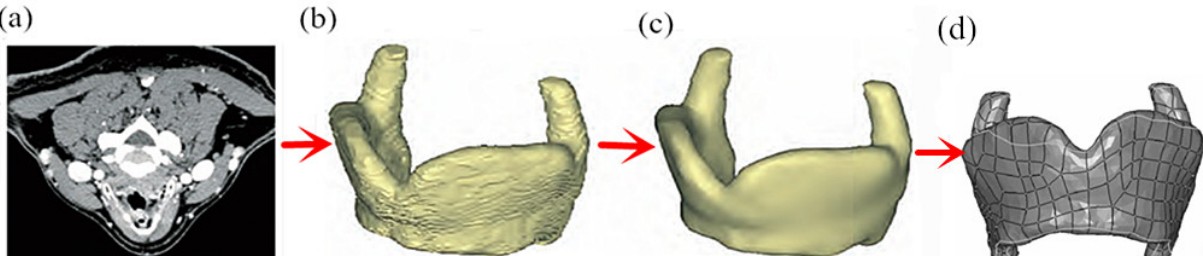

**Figure 3.** Design process of solid thyroid cartilage: (**a**) CT images; (**b**) reconstructed model; (**c**) model after surface treatment; (**d**) materialized model.

### 3.2. Parametric Modeling of Total Resection of Thyroid Cartilage Prosthesis

During use of a solid prosthesis, "stress shielding" often occurs because the hardness is greater than the peripheral bone tissue, so growth of the peripheral tissue is not stimulated. In addition, a solid prosthesis is unable to grow into the peripheral bone tissue due to its lack of pores, which can cause loosening. Therefore, lower requirements of mechanical properties and wear resistance of the repaired parts would allow some parts to be made as porous structures. According to the structural characteristics, a single-layer grid structure or a multi-layer grid structure could be used for the total resection of thyroid cartilage prosthesis.

### 3.2.1. Single-Layer Grid Structure of Thyroid Cartilage

Single-layer grid structure is widely used because of its simple structure, simple modeling process, and the lightweight nature of the material for implantation. The specific modeling process is as follows:

(1) The 3D reconstructed thyroid cartilage implant model was imported into Rhinoceros v5 software to reconstruct a solid model based on the original model. Fixed pores were established at the edges of both sides, and the outer surface of the model was extracted and reconstructed, as shown in Figure 4a.

(2) In the Rhinoceros software, the surface battery of the grasshopper plug-in was used to read the external surface of the reconstructed model, the battery chart program was written, the grid lines were generated by the Braced Grid 1-D structure plug-in, the grid line density was adjusted to meet the implant implantation requirements, and the point cloud information was deleted, as shown in Figure 4b.

(3) The gridlines were read in by the Curve parameter battery of the Grasshopper plug-in, and then surface tubes were generated as lines through the pipe battery. The diameter of the surface tubes was adjusted to meet the implantation requirements. The surface tubes were generated in the solid model through the BREP join battery, and then incorporated into the Rhinoceros software, as shown in Figure 4c. It can be seen from Figure 4c that the parametric modeling method can effectively realize joint modeling of truss porous structure and solid structure, with good modeling effects, uniform pore distribution, good connectivity, good overlapping between pillars and pore size that meets the requirements of biocompatibility, and an amount of overlap suspension between pillars that meets the requirements of SLM manufacturing, with high bearing capacity.

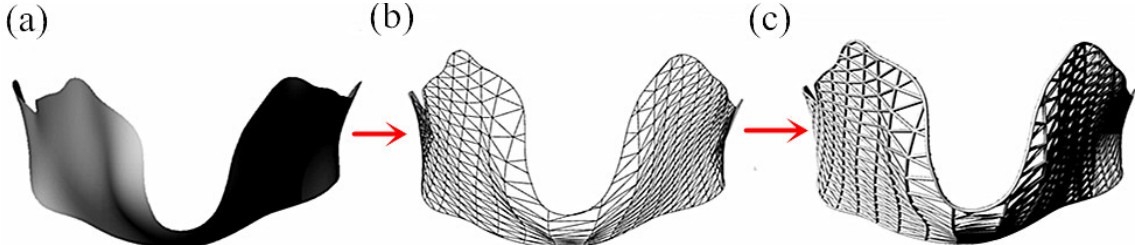

**Figure 4.** Design process of single-layer grid structure for thyroid cartilage prosthesis after total resection: (**a**) surface model; (**b**) line model; (**c**) volume model.

### 3.2.2. Multi-Layer Thyroid Cartilage Structure

The truss structure is not only light in weight but also has good mechanical properties of tension and compression, allowing its use in the design of thyroid cartilage prosthesis. To further improve the biocompatibility and mechanical properties of the thyroid cartilage prosthesis, the truss network can be appropriately thickened to make a double-layer or multi-layer truss network. The specific modeling process is as follows:

(1) The 3D-reconstructed thyroid cartilage implant model was imported into Rhinoceros 5 software to reconstruct a solid model based on the original model, fixed pores were established at the edges of both sides, and the outer surface of the model was extracted and reconstructed, as shown in Figure 5a.

(2) In the Rhinoceros software, the surface battery of the grasshopper plug-in was used to read the external surface of the reconstructed model, the battery chart program was written, the truss structure lines were generated by the Space Truss Structure plug-in, and the grid line density and the distance between upper and lower chords was adjusted to meet the implant implantation requirements, and baked into the software to delete the point cloud information, as shown in Figure 5b.

(3) The gridlines were read in by the Curve parameter battery of the Grasshopper plug-in, and then the surface tubes were generated as lines through the pipe battery. The diameter of the surface tubes was adjusted to meet the implantation requirements. The surface tubes were generated in the solid model through the Brep join battery, and baked into the Rhinoceros software, as shown in Figure 5c. It can be seen from Figure 5c that in addition to the better mechanical properties of the single-layer grid structure, the multi-layer grid structure of thyroid cartilage with parametric modeling method is more conducive for growth of bone tissue and offers better biocompatibility due to the gradient change in the top grid and bottom layer of the truss structure.

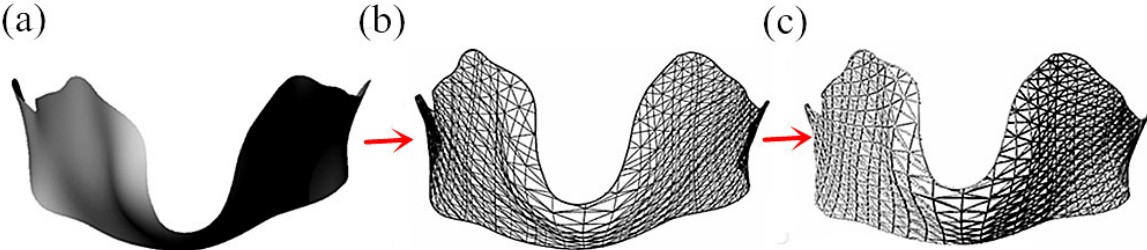

**Figure 5.** Design process of multi-layer grid structure for thyroid cartilage prosthesis after total resection: (**a**) Surface model; (**b**) Line model; (**c**) Volume model.

### 3.3. Parametric Modeling of Prosthesis of Thyroid Cartilage with Partial Resection

The implant was constructed by a parametric modeling method of the total excision of the thyroid cartilage with partial resection. If the affected site is small, partial resection can be used to manufacture the parametric thyroid cartilage prosthesis. The specific process is as follows:

(1) The doctor images the affected site, as shown in Figure 6a, and then performs three-dimensional reconstruction of the thyroid cartilage prosthesis as described in Section 3.1.

(2) According to the size of the area diagnosed and the condition of the patient, the doctor determines the scope of the osteotomy, draws the osteotomy line, and carries out simulated osteotomy, as shown in Figure 6b.

(3) To facilitate fixation, reduce weight, and improve matching, the Rhinoceros software is used to fix the contact parts of the thyroid cartilage and unaffected parts using a circular arc-shaped clamping groove. The interface is limited to five degrees of freedom, and upward movement can be limited by screws, which will greatly reduce the number of screws and fixed plates. To further reduce weight, pores can be made on the circular arc-shaped clamping groove, as shown in Figure 6c.

(4) According to the requirements of biocompatibility and the mechanical properties of porous implants, the truss line is established using the grasshopper plug-in of the Rhinoceros software; then, the curved surface flows to the site where the thyroid cartilage is removed, and the redundant curved surface is deleted, as shown in Figure 6d.

(5) The gridlines are read in by the Curve parameter battery of the Grasshopper plug-in, and then the surface tubes are generated as lines through the pipe battery. The diameter of the surface tubes is adjusted to meet the implantation requirements. The surface tubes are generated into the solid model through the Brep join battery, and baked into Rhinoceros software, as shown in Figure 6e. It can be seen from Figure 6e that partial resection of the thyroid cartilage prosthesis completed by the parametric modeling method shows good modeling effects, but because the solid truss structure is generated by a truss line, there will be convex points formed on the concave arc groove and the external surface of the prosthesis. This will affect the surface quality of the prosthesis and the degree of matching after implantation, so further improvement is needed.

(6) After closure and reconstruction of the surface pores of the thyroid cartilage prosthesis, the convex points on the surface were eliminated by Boolean operation. The results after elimination of the convex points are shown in Figure 7. The surfaces of the concave arc groove and the concha cartilage were smooth after eliminating the external convex points, which greatly improved the fit between the implant and the unaffected site. After using the Rhinoceros software to design and complete the three-dimensional model of thyroid cartilage prosthesis, before printing, a series of model checks was performed for the designed model. The thyroid cartilage repair model and the thyroid cartilage model of the unaffected part were then imported into the SolidWorks software for a solid check as follows: click Tools → evaluation → check option, select all the entities and curved surfaces, and then select the invalid surface and invalid sideline in the search to optimize and improve the design.

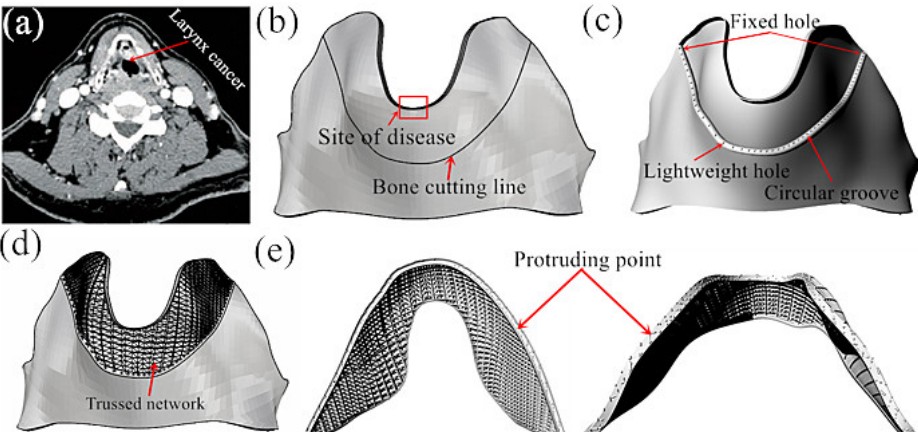

**Figure 6.** The design process of total removal of multi-layer mesh thyroid cartilage prosthesis: (**a**) CT model; (**b**) simulated osteotomy; (**c**) opening position of fixed hole; (**d**) line model; (**e**) volume model.

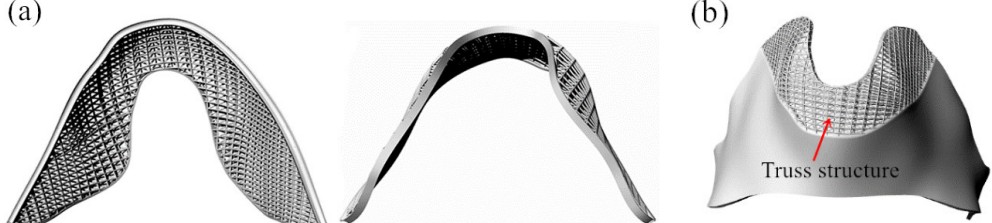

**Figure 7.** Total removal of multi-layer mesh of thyroid cartilage prosthesis after removal of convex points. (**a**) Convex points elimination effect; (**b**) Matching effect.

(7) A matching check of thyroid cartilage prosthesis model was performed in the Solid-Works 2019 software. This was done as follows: click Tools → evaluation → interference check option; select the combination of thyroid cartilage prosthesis in the selected parts, and then conduct interference calculation. The result shows good matching of the two models without interference, suggesting good matching of the printed model.

*3.4. Research on the Technology of SLM Molding Thyroid Cartilage Prosthesis*

Based on a previous report of the use of SLM for the molding of parts [22], the prominentia laryngea of the thyroid cartilage prosthesis was positioned vertically to the basal plate, and the external contour of the prominentia laryngea was supported by a block and a heat conducting column in the thyroid cartilage prosthesis model, as shown in Figure 8. This placement and support adding method can avoid the requirement for the addition of support to the arc-shaped groove and the warping deformation of the parts, and thus improve the matching and molding quality of the prosthesis.

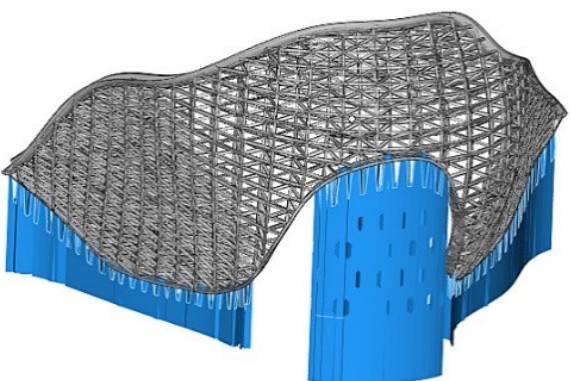

**Figure 8.** Thyroid cartilage prosthesis with data processing.

The designed thyroid cartilage prosthesis formed by SLM is shown in Figure 9a. The thyroid cartilage prosthesis formed by SLM lacks warping deformation, and has a high surface finish and less powder adhesion, suggesting high shape accuracy and internal quality sufficient to meet the use requirements. As shown in Figure 9b, the microstructure of the SLM molding thyroid cartilage prosthesis reveals good overlap between the porous pillars of the thyroid cartilage, good connectivity between pores, less powder adhesion on the surface, and no obvious nodulation between the porous pillars.

*3.5. Matching Test of SLM Molding Thyroid Cartilage Prosthesis*

To evaluate the effects of potential errors in the size and shape accuracy, we assembled the thyroid cartilage prosthesis and the Z-603S 3D printed parts of the unresected part to check the matching. The assembly results are shown in Figure 10. The corrected SLM molding thyroid cartilage prosthesis fits well with the arc-shaped groove of the unresected part of the thyroid cartilage and the convex part of the thyroid cartilage, realizing a smooth transition between the prosthesis model and the unresected part. Thus, the resulting thyroid cartilage prosthesis can be used directly after heat treatment and anodization.

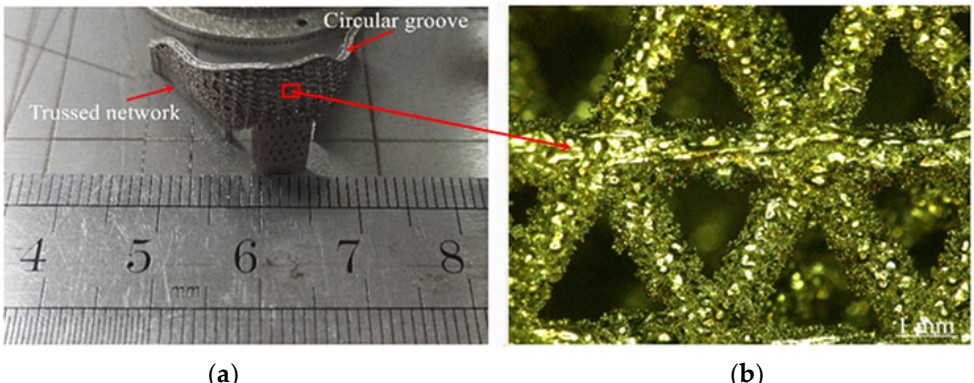

(**a**)                    (**b**)

**Figure 9.** Thyroid cartilage prosthesis made by SLM: (**a**) SLM formed parts; (**b**) microstructure.

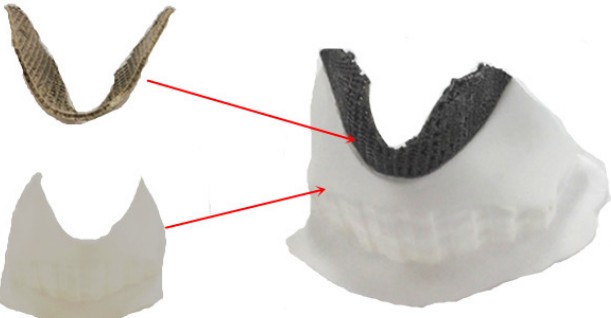

**Figure 10.** Matching test chart of thyroid cartilage prosthesis formed by SLM.

## 4. Conclusions

1.  A thyroid cartilage prosthesis was designed using a parameterized method with total or partial resection. The modeling effect was good, with good matching to unaffected sites, good overlap between pillars, appropriate pore size that meets the requirements of biocompatibility, and good overlap suspension between pillars to meet the requirements of SLM manufacturing.
2.  The thyroid cartilage prosthesis prepared by SLM molding has better overlap between porous pillars, better connectivity between pores, less powder adhesion on the surface, and no obvious nodulation between porous pillars. The resulting prosthesis can be directly used after simple sandblasting and polishing.
3.  The resulting thyroid cartilage prosthesis fits well with the arc-shaped groove of the unresected part of the thyroid cartilage and the convex part of the thyroid cartilage, providing a smooth transition between the prosthesis model and the unresected part so that the prosthesis can be directly used after heat treatment and anodizing.

Subsequent studies should further explore the design and molding of thyroid cartilage prosthesis, such as the influence of the method on adding support and the effects of the SLM molding process parameters on the molding quality of the implant. This work provides the foundation for the direct manufacture of a thyroid cartilage prosthesis by SLM.

**Author Contributions:** G.Z. and J.L. completed the design of implants. C.Z. and A.W. completed the manufacturing and analysis of implants. All authors have read and agreed to the published version of the manuscript.

**Funding:** The study was funded by the Henan Provincial Science and Technology Project (Grant No. 212102310859) and the Key Scientific Research Projects of Colleges and Universities in Henan Province (Grant No. 22A460006).

**Institutional Review Board Statement:** Not applicable.

**Informed Consent Statement:** Not applicable.

**Data Availability Statement:** Data sharing is not applicable to this article.

**Acknowledgments:** The carrying out of analysis for this work was also supported by the Analytical and Testing Center of Zhoukou Normal University.

**Conflicts of Interest:** Data sharing is not applicable to this article.

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
