# Peer review of "Design and Molding of Thyroid Cartilage Prosthesis Based on 3D Printing Technology"

_coatings, doi:10.3390/coatings12030336_

Round 1
Reviewer 1 Report
The authors have nicely explained and attempted the real time manufacturing problem for bio-medical application. I have only one submission about the fact that, currently how the problem is tackled. The readers would like to read a paragraph about the same with some appropriate citations.
Author Response
We have performed the comprehensive modifications to the writing problems of this paper. (in red)
We have carefully read the related articles and cited them in the introduction.. (in red)
- Tian Zhen, Yu Jianjun, Zhou Xiao, Chen Xing, Zuo Liang, Gao Shuichao, Cai Xu, song Bo, Liu Xuekui. Application of digital technique combined with 3D printing technique in reconstruction of thyroid cartilage in partial laryngectomy. Chinese Otorhinolaryngology Head and neck surgery 27.1 (2020): 20-24.
- Zhang Longcheng, Hu Wanqing, Lin Wenbiao, Cao Gaoxiang, Huang Haibo, Xia Wei, Lu Mengyi, Wei Ganguan. Experimental study on repair of thyroid cartilage defect with rhBMP-2 / Cha. Journal of Huazhong University of science and Technology (Medical Edition) 43.3 (2014): 316-320.
- Hallak B, Von Wihl S, Boselie F, et al. Repair of displaced thyroid cartilage fracture using miniplate osteosynthesis. BMJ Case Reports CP 11.1 (2018): e226677.
Reviewer 2 Report
In the manuscript entitled "Design and Molding of Thyroid Cartilage Prosthesis Based on 3D Printing Technology" the authors are reporting a the process used to design a thyroid cartilage prosthesis using the 3D printing technology. The paper has potential but improvements are needed to improve the manuscript.
In Figure 1 the two images should be named a) and b) and in the legend both have to be explained in detail.
For Figure 3-6 in the legend it should be explained what the component images represent.
In figure 9 the authors need to name the two figures and explain in the legend what is in each one.
Based on the changes in the figure legends the text should also be adapted.
The manuscript can be considered if the changes are made.
Author Response
Point 1: In the manuscript entitled "Design and Molding of Thyroid Cartilage Prosthesis Based on 3D Printing Technology" the authors are reporting a the process used to design a thyroid cartilage prosthesis using the 3D printing technology. The paper has potential but improvements are needed to improve the manuscript.
In Figure 1 the two images should be named a) and b) and in the legend both have to be explained in detail.
For Figure 3-6 in the legend it should be explained what the component images represent.
In figure 9 the authors need to name the two figures and explain in the legend what is in each one.
Based on the changes in the figure legends the text should also be adapted.
The manuscript can be considered if the changes are made.
Response 1: We have named two images in Fig. 1 as (a) and (b) and explained them, then named the pictures in Figures 3-6, 9 and explained separately. (in red).
Reviewer 3 Report
Dear Authors,
Thank you for interesting paper to review. The 3D printing technology belongs to the very interesting and promising processes of formation high-precision modelled part espectially for the biomedical application.
After reviewing your work, I have the following comments and questions.
1. Why did you used Ti6Al4V alloy as studied material?
Of course from the one side this alloy is characterized by excellent mechanical properties and corrosion resistance, however its alloying additions are unwanted for human body. Because aluminum affects aluminosis, while the vanadium is strong toxic element.
Please, explain it.
2. The paragraphs number 3.1-3.3 describe the design of solid thyroid cartilage prosthesis, multi-layer thyroid crtilage structure, parametric modelling. However, from my point of view, these are not the result of investigations but the description of methods modeling. Therefore, I think that these parts of paper should be moved to section Materials and Methods.
3. Par 3.4 - line 296/297. What does "high surface finish" mean? What was the surface roughness of obtained model? Did you performed such measurements?
4. There is no analysis of mechanical properties of obtained thyroid cartilage prosthesis. I think that it is important aspect in terms of potential application.
5. There is no images of cross section of produced model. I think that the analysis of microstrucutre, morphology/porosity should be added.
6. Conclusions - Line 324/325. You wrote that the obtained prosthesis can be used after simple sandlasting and polishing. The first process allows to obtain the rough surface of part, while the result of the polishing is smoothing surface. So, I do not understand, what the aim of both surface treatment. The surface properties like roughness, porosity of near-surface area are very important for such application. The appropriate surface properties provide the good connection of the implant with the tissue and acceleration of healing processes. Please, explain it.
Thank you very much for answers in advance.
Best regards,
Reviewer
Author Response
Point 1: Why did you used Ti6Al4V alloy as studied material?
Of course from the one side this alloy is characterized by excellent mechanical properties and corrosion resistance, however its alloying additions are unwanted for human body. Because aluminum affects aluminosis, while the vanadium is strong toxic element.
Please, explain it.
Response 1: Ti6Al4V is the most widely used medical material with the best comprehensive performance. Due to the consensus in relevant fields, there is no too much explanation.
Point 2: The paragraphs number 3.1-3.3 describe the design of solid thyroid cartilage prosthesis, multi-layer thyroid crtilage structure, parametric modelling. However, from my point of view, these are not the result of investigations but the description of methods modeling. Therefore, I think that these parts of paper should be moved to section Materials and Methods.
Response 2: The purpose of this writing is mainly to correspond to the topic. This paper mainly studies the design and manufacturing methods.
Point 3: Par 3.4 - line 296/297. What does "high surface finish" mean? What was the surface roughness of obtained model? Did you performed such measurements?
Response3: The surface brightness is the visual observation result of the surface quality of parts. We have added that the measurement result of coarseness is 11 μ m。
Point 4: There is no analysis of mechanical properties of obtained thyroid cartilage prosthesis. I think that it is important aspect in terms of potential application.
Response4: We have not yet discussed the overall mechanical performance of 3D printing parts due to the difficulties in testing, but we have discussed the performance of similar porous structural parts of 3D printing in in other papers.
Zhang Guoqing, Yang Yongqiang, song Changhui, et al Design and properties of CoCrMo porous structure formed by laser selective melting [J] China laser, 2015 (11): 10
Wang Xiaolong, Xiao Zhiyu, Zhang Guoqing, et al Effect of tilt angle on laser selective melting of Ti6Al4V alloy [J] Powder metallurgy materials science and engineering, 2016, 021 (003): 376-382
Point 5: There is no images of cross section of produced model. I think that the analysis of microstrucutre, morphology/porosity should be added.
Response5: Due to the complex structure of the designed thyroid cartilage, it is inconvenient to test, but we have carried out relevant analysis in other papers.
Guoqing Z , Yongqiang Y , Lin Hui…. Study on the Quality and Performance of CoCrMo Alloy Parts Manufactured by Selective Laser Melting[J]. Journal of Materials Engineering and Performance, 2017, 26(6):1-9.
Point 6: Conclusions - Line 324/325. You wrote that the obtained prosthesis can be used after simple sandlasting and polishing. The first process allows to obtain the rough surface of part, while the result of the polishing is smoothing surface. So, I do not understand, what the aim of both surface treatment. The surface properties like roughness, porosity of near-surface area are very important for such application. The appropriate surface properties provide the good connection of the implant with the tissue and acceleration of healing processes. Please, explain it.
Response6: We have deleted some expressions that lack preciseness.
Reviewer 4 Report
The present Manuscript is aimed to solving the important problem of thyroid cartilage prosthesis design, attracting the novel 3D printing technique. Different models of prosthesis are discussed. However, I consider it necessary to make following comments:
- Information about material used for prosthesis fabrication should be added to Abstract.
- Introduction section should be revised significantly. There is a lack of references which prove a lot of statements and there is no information about materials used in the considering area.
- Line 115: “scanning distance” should be changed for common “hatch spacing”. "80um” should be “80 µm”.
- Line 114-117: Why only one SLM regime was used? Is it an optimum, how was it chosen?
- The observation of various approaches of thyroid cartilage prosthesis modeling is presented in Sections 3.1-3.3 without their comparing.
- There is no discussion, modeling or experimental testing of designed prosthesis mechanical properties is the Manuscript. It should be at least discussed theoretically, but it will be better to perform the related study.
- Authors almost did not discuss biocompatibility issue of the prosthesis material, which is one of the most important in prosthesis design. There is also no experimental study concerning it.
- The style of the Manuscript doesn't seem scientific in some cases. Namely, the text is full of phrases like “effect was good”, “good matching”, etc. All the statements should be proved theoretically or experimentally.
- The conclusion that “The resulting prosthesis can be directly used after simple sandblasting and polishing” seem premature in light of the above, especially since authors suggest “Subsequent studies are needed to further explore the design and molding of thyroid cartilage prosthesis…” below.
Thereby, deeper analysis of the literature data, more detailed discussion of the results, additional theoretical and/or experimental investigations and style corrections to improve scientific soundness are required.
Author Response
Point 1: Information about material used for prosthesis fabrication should be added to Abstract.
Response 1: We have appended the formed material in the abstract as TC4. (in red)
Point 2: Introduction section should be revised significantly. There is a lack of references which prove a lot of statements and there is no information about materials used in the considering area.
Response 2: We have supplemented relevant references in the introduction. (in red)
- Tian Zhen, Yu Jianjun, Zhou Xiao, Chen Xing, Zuo Liang, Gao Shuichao, Cai Xu, song Bo, Liu Xuekui. Application of digital technique combined with 3D printing technique in reconstruction of thyroid cartilage in partial laryngectomy. Chinese Otorhinolaryngology Head and neck surgery 27.1 (2020): 20-24.
- Zhang Longcheng, Hu Wanqing, Lin Wenbiao, Cao Gaoxiang, Huang Haibo, Xia Wei, Lu Mengyi, Wei Ganguan. Experimental study on repair of thyroid cartilage defect with rhBMP-2 / Cha. Journal of Huazhong University of science and Technology (Medical Edition) 43.3 (2014): 316-320.
- Hallak B, Von Wihl S, Boselie F, et al. Repair of displaced thyroid cartilage fracture using miniplate osteosynthesis. BMJ Case Reports CP 11.1 (2018): e226677.
Point 3: Line 115: “scanning distance” should be changed for common “hatch spacing”. "80um” should be “80 µm”.
Response3: We have changed the scanning spacing to 80 μm. (in red)
Point 4: Line 114-117: Why only one SLM regime was used? Is it an optimum, how was it chosen?
Response 4: To cope with such complex metal grid structure, SLM technology manufacturing is a preferable choice currently.
Point 5: The observation of various approaches of thyroid cartilage prosthesis modeling is presented in Sections 3.1-3.3 without their comparing.
Response5: We have introduced the disadvantages of the above modeling method in section 3.3 of the paper.
(It can be seen from Figure 6 (e) that partial resection of the thyroid cartilage prosthesis completed by the parametric modeling method showed good modeling effect, but because the solid truss structure is generated by a truss line, there will be convex points formed on the concave arc groove and the external surface of the prosthesis. This will affect the surface quality of the prosthesis and the degree of matching after implantation, so further improvement is needed.
)。
Point 6: There is no discussion, modeling or experimental testing of designed prosthesis mechanical properties is the Manuscript. It should be at least discussed theoretically, but it will be better to perform the related study.
Response 6: We have not yet discussed the overall mechanical performance of 3D printing parts due to the difficulties in testing, but we have discussed the performance of similar porous structural parts of 3D printing in in other papers.
Zhang Guoqing, Yang Yongqiang, song Changhui, et al Design and properties of CoCrMo porous structure formed by laser selective melting [J] China laser, 2015 (11): 10
Wang Xiaolong, Xiao Zhiyu, Zhang Guoqing, et al Effect of tilt angle on laser selective melting of Ti6Al4V alloy [J] Powder metallurgy materials science and engineering, 2016, 021 (003): 376-382
Point 7: Authors almost did not discuss biocompatibility issue of the prosthesis material, which is one of the most important in prosthesis design. There is also no experimental study concerning it.
Response 7: We have analyzed similar structures in other papers.
Guoqing Z , Junxin L , Jin L , et al. Simulation Analysis and Performance Study of CoCrMo Porous Structure Manufactured by Selective Laser Melting[J]. Journal of Materials Engineering & Performance, 2018.
Point 8: The style of the Manuscript doesn't seem scientific in some cases. Namely, the text is full of phrases like “effect was good”, “good matching”, etc. All the statements should be proved theoretically or experimentally.
Response 8: The satisfactory effect stated in the paper refers to the surface quality of SLM manufactured parts, and it is concluded from the assembly experiment on the 3D printed plastic model the matching is good.
Point 9: The conclusion that “The resulting prosthesis can be directly used after simple sandblasting and polishing” seem premature in light of the above, especially since authors suggest “Subsequent studies are needed to further explore the design and molding of thyroid cartilage prosthesis…” below.
Response9: We have deleted some expressions that lack preciseness.
Round 2
Reviewer 2 Report
The authors have made changes to the manuscript. The quality improved and the manuscript can be accepted for publication.
Author Response
Thank you!
Reviewer 3 Report
Dear Authors,
Thank you for all responses and explains.
I accept the reviewed paper in the presented form.
Best regards,
Reviewer
Author Response
Thank you!Reviewer 4 Report
Authors made some significant corrections of the manuscript, but I am still not satisfied with some responses.
Point 1. – OK.
Point 2. Introduction section becomes better and more logical.
Point 3. “scanning distance” was not changed for “hatch spacing” in spite of authors response.
Point 4. My comment was not about choice of method (SLM or not) but about specific regime of the SLM process. Why “processing laser power was 180W, scanning speed was 500 mm / s, the scanning distance was 80μm” were chosen? Is it based on previous research or literature data? References are needed in both cases.
Point 5. – OK.
Point 6. If some investigations were performed in other papers corresponding discussion with references should be added to the manuscript.
Point 7. The same. If some investigations were performed in other papers corresponding discussion with references should be added to the manuscript.
Point 8. There are a lot of scientific different methods to analyze surface quality of SLMed samples and parts, not just “matching”. This statement does not sound scientific. Moreover, no cross-section analysis of microstructure was performed, that decrease the valuable of obtained results.
Point 9. OK.
Author Response
Point 3: “scanning distance” was not changed for “hatch spacing” in spite of authors response.
Response3: We have changed “scanning distance” to “hatch spacing”. (in red)
Point 4: My comment was not about choice of method (SLM or not) but about specific regime of the SLM process. Why “processing laser power was 180W, scanning speed was 500 mm / s, the scanning distance was 80μm” were chosen? Is it based on previous research or literature data? References are needed in both cases.
Response4:Based on the peer research of the research group, we have supplemented relevant references.
Point 6: If some investigations were performed in other papers corresponding discussion with references should be added to the manuscript.
Response 6: We have not yet discussed the overall mechanical performance of 3D printing parts due to the difficulties in testing, but we have discussed the performance of similar porous structural parts of 3D printing in in other papers.
Zhang Guoqing, Yang Yongqiang, song Changhui, et al Design and properties of CoCrMo porous structure formed by laser selective melting [J] China laser, 2015 (11): 10
Wang Xiaolong, Xiao Zhiyu, Zhang Guoqing, et al Effect of tilt angle on laser selective melting of Ti6Al4V alloy [J] Powder metallurgy materials science and engineering, 2016, 021 (003): 376-382
Point 7: The same. If some investigations were performed in other papers corresponding discussion with references should be added to the manuscript.
Response 7: We have analyzed similar structures in other papers.
Guoqing Z , Junxin L , Jin L , et al. Simulation Analysis and Performance Study of CoCrMo Porous Structure Manufactured by Selective Laser Melting[J]. Journal of Materials Engineering & Performance, 2018.
Point 8: There are a lot of scientific different methods to analyze surface quality of SLMed samples and parts, not just “matching”. This statement does not sound scientific. Moreover, no cross-section analysis of microstructure was performed, that decrease the valuable of obtained results.
Response 8: Due to the mature manufacturing process selected, the quality of the parts themselves is basically no problem. Thyroid cartilage is easy to deform due to its large size, so matching is particularly important